# Explaining COVID-19 Vaccine Rejection Using Social Cognitive Theory in Qassim, Saudi Arabia

**DOI:** 10.3390/vaccines9111304

**Published:** 2021-11-09

**Authors:** Aseel Ali AlSaeed, Unaib Rabbani

**Affiliations:** Family Medicine Academy, Qassim Health Cluster, Buraidah 52385, Saudi Arabia; rabbaniunaib@gmail.com

**Keywords:** COVID-19, vaccine hesitancy, barriers to vaccine, social cognitive theory, Saudi Arabia

## Abstract

Acceptance of COVID-19 vaccines needs a health promotion approach to address various social, environmental and personal factors leading to vaccine hesitancy. We assessed the vaccine hesitancy rate and applied social cognitive theory (SCT) to understand COVID-19 vaccine rejection in Qassim, Saudi Arabia. A cross-sectional study was conducted among visitors of 10 randomly selected primary health care centers in Buraidah, Saudi Arabia. Data was collected by a self-administrated questionnaire. The variables were grouped into six constructs of SCT. Logistic regression was used to assess the predictors of vaccine rejection. Out of 486 participants included in the study, 30.5% rejected the vaccine. The most common reason for vaccine rejection was uncertainty about the vaccine’s effectiveness (78%). Among various constructs of SCT, reciprocal determinism (nationality, income and suffering from COVID-19 infection), behavioral capability (knowledge about vaccine safety), self-efficacy (registered for vaccine), and observational learning (getting the vaccine after friends and family members) were significant predictors. Expectation and reinforcement constructs did not show significant association. There was high vaccine rejection in Qassim, KSA. This calls for further improving the mass education strategies. Social cognitive theory can be used to predict vaccine rejection and to develop strategies to increase the utilization of COVID-19 vaccines in Saudi Arabia.

## 1. Introduction

Millions of people worldwide have been infected and/or died due to severe acute respiratory syndrome coronavirus 2 (SARS-CoV-2) or COVID-19 infection since the start of the pandemic [1]. Referring to the number of cases and deaths, this pandemic is a serious threat to global public health systems [2].

Developing an effective and safe vaccine was a promising hope since the early days of the pandemic. Vaccination is considered one of the most effective strategies to control the pandemic along with other measures such as social distancing, masks, and the use of sanitizers [3]. However, the availability of a vaccine is not enough to control the pandemic. Vaccine acceptance is very important to assess, as it reflects the overall perception of disease risk in the population [4]. A number of factors associated with rejection have been reported in various studies such as vaccine safety and efficacy, lack of trust in health system, misconception, and misinformation. All these factors can lead to low vaccination rates and jeopardize public health [5,6].

Globally, vaccine acceptance has been widely studied and has shown wide variations. A global survey that included participants from 19 countries reported that about 75% of the respondent were “very to somewhat” likely to accept COVID-19 vaccination [7]. A systematic review that included studies from 33 countries reported acceptance rates to be highest (more than 90%) in Ecuador, Malaysia, Indonesia, and China. On the other hand, Kuwait (23.6%) and Jordan (28.4%) showed the lowest acceptance rates [8]. Two different surveys that included participants from more than 19 Arab countries reported vaccine acceptance to be between 62% and 58.5% [9,10]. Similarly, a number of studies have been published from Saudi Arabia, which reported vaccine acceptance rates to range from 48% to 72% [5,6,11,12,13]. The common barriers reported among the Saudi population include concerns about the safety and efficacy of vaccines, vaccines being a conspiracy, and a perceived low risk of COVID-19 infection [6,11,12]. Studies have also reported various factors associated with vaccine uptake which include fear of infection, level of trust of vaccines, level of trust in the health care system, age, gender, marital status, education, presence of chronic diseases, and previous influenza vaccine uptake [4,10,12,13].

Vaccine acceptance is very important to assess, as the success of any vaccination programs depend on the public’s willingness to take it. There are many theories from social sciences that have been used to understand health behaviors and promotion of healthy behaviors. Some of these theories are used at an individual level while others are used at group and community levels. Vaccine acceptance, while a personal decision, is influenced by various environmental and social factors. Previous studies have used these theories to explain the vaccine acceptance rate among individuals and groups. Studies on influenza, human papilloma virus, and hepatitis B vaccinations have used such theories to explain the factors associated with acceptance of these vaccines [14,15,16]. Recently, a number of studies have used these theories to understand COVID-19 vaccination behaviors. Most of these studies have mainly used the health belief model (HBM) [17,18,19,20,21] and only few have used social cognitive theory (SCT) [22]. Nonetheless, these studies have been able to predict the COVID-19 vaccine utilization rate at an individual level. However, COVID-19 vaccination is also affected by various factors outside of an individual such as behaviors of others towards vaccination, governmental policies regarding vaccination, and media exposure.

The Saudi government took various steps to curb the spread of COVID-19, which included lockdowns, travel restrictions, use of masks and hand sanitizers, and mass education about prevention of disease. Furthermore, in the month of April 2021, the government made it mandatory to be vaccinated to visit the Holy sites of Makkah and Madinah. Later, the General authority for civil aviation (GACA) announced that starting from 17 May, travelling outside the Kingdom will only be allowed for vaccinated persons. Despite all these measures widespread vaccine rejection has been reported.

Vaccination is critical for controlling the current COVID-19 pandemic. Resistance towards vaccination is a threat to global public health. It is therefore necessary to address this issue through a health promotion approach. SCT, an interpersonal framework, can be helpful in explaining the personal and social factors influencing vaccine rejection and help develop strategies to improve the utilization of COVID-19 vaccines in the general population. This study therefore aimed to assess COVID-19 vaccine acceptance and apply SCT to explain the rejection of vaccines and its barriers among primary health care (PHC) center attendees in the Qassim region of Saudi Arabia.

## 2. Materials and Methods

### 2.1. Study Design and Setting

A cross-sectional study was conducted among the attendees of selected primary health care centers in Buraidah from February 2021 to June 2021. Buraidah is the capital of Qassim region with an estimated population of 693,515 [23].

### 2.2. Sample Size

The sample size for the study was calculated using WHO software for sample size determination in health studies. The sample size was measured using 44.7% prevalence of COVID-19 vaccination acceptance [6]. At 95% confidence level and 5% bound on error, the maximum calculated sample size was 380 participants.

### 2.3. Sampling Technique and Procedure

There are 40 primary health care centers in Buraidah. Out of these, 10 centers were selected by a simple random sampling. Within each selected PHC, a total of 40–50 participants were selected consecutively by convenience sampling. Males and females ≥18 years of age and citizens and residents were eligible to participate. Those who had vaccine contraindication according to Saudi Ministry of Health (MoH) (having a history of a severe allergic reaction, pregnant women, those planning to conceive in next 2 months, and those within 90 days of COVID-19 infection at the time of data collection) were excluded from the study. The purpose of the study and its objectives were explained to the patients who were in the waiting rooms. Participants were assessed for eligibility. Those meeting eligibility criteria were invited to participate in the study.

### 2.4. Data Collection Tool

Data was collected by a structured questionnaire in Arabic. The questionnaire consisted of five domains. The domains aimed to collect data on socio-demographic variables, awareness about COVID-19 and its vaccine, vaccine acceptance, reasons for not opting for COVID-19 vaccination, and sources of COVID-19 vaccine information. The questionnaire was developed after a review of the literature [3,4,5,6,24]. Study variables were identified to meet the objectives of the study. The draft questionnaire was reviewed by research faculty in the program and finalized.

The items were organized into six constructs of SCT. Socio-demographic characteristics and history of COVID-19 infection in family or individual were grouped in reciprocal determinism. Knowledge and practices related to COVID-19 infection prevention were kept under behavioral capabilities. Risk and severity of infection were included in expectations. Intention and registration to receive COVID-19 vaccine constituted self-efficacy. Observational learning was assessed by the question “I will get COVID-19 vaccine only if my friends or other family members get it first” while reinforcement construct included a vaccine requirement by an employer and enforcement by the government.

The questionnaire was translated into Arabic first and then back-translated to English by an independent translator for validation of the Arabic translation of the questionnaire. A pilot study was conducted to ensure the clarity of questions, applicability to the participants, and to identify if there was any problem that could have impeded the data collection process. After pilot testing, the questionnaire was modified according to the observations during testing.

### 2.5. Data Collection Procedure

Data was collected by trained data collectors who were either doctors or nurses. First, the study purpose and procedure were explained and informed consent was obtained. Printed questionnaires were provided to the participants. Participants were required to fill out the form by themselves; data collectors were available to assist if any clarification was required.

### 2.6. Data Entry and Analysis

Data was entered and analyzed using SPSS version 23. Descriptive analysis was carried out in the form of frequencies and percentage for categorical variables while mean and standard deviation (SD) were calculated for continuous variables. Logistic regression analysis was used to assess the predictors of vaccine rejection. Variables that had a *p*-values of ≤0.2 in the univariate analysis were included in the multivariate models. Variables in the final model were retained based on their significance and effects on −2 log likelihood ratio. Adjusted and unadjusted odds ratio along with associated 95% confidence interval were calculated. A *p*-value < 0.05 was considered significant for all inferential analysis.

Ethical approval of the study was taken from the Qassim Regional Bioethics Committee. Informed consent was obtained from all participants.

## 3. Results

A total of 486 participants completed the survey questionnaire of which 54% were males. Fifty-four percent were married and a majority (90.5%) were Saudi nationals. Approximately one third (34.8%) had a high school or lower educational level, whereas 65.2% had a diploma or higher. Regarding health status, 84.2% did not have any chronic disease and 28.7% had received an influenza vaccine during the previous year. Around 16% reported having suffered from COVID-19 while 37.3% had one of their family members infected with the disease (Table 1).

Table 2 shows participants’ awareness and behaviors towards COVID-19. It was found that 68.2% believed COVID-19 could lead to death and 91.9% reported that COVID-19 spreads by close contact with infected people. A majority (97.7%) were aware of the COVID-19 vaccine and 61% believed it to be safe. Half (50.9%) of the respondents had already registered for COVID-19 vaccination and 28% were waiting for their family or friends to be vaccinated before they would be vaccinated.

Regarding the sources of information for COVID-19, the most common was TV/radio (67.9%) followed by social media (67.70%). Other sources of information included the Ministry of Health (64.40%), friends and family members (54.30%), health care providers (53.50%), and newspapers (26.50%).

The vaccine rejection rate was found to be 30.50%. A significant difference was found in the vaccine rejection rates between pre- and post-enforcement periods (59.3% vs. 40.7%; *p* value < 0.001).

The most commonly reported reason behind vaccine rejection was the belief that vaccines may not be effective (78.4%) and the least common reason was that vaccines are a conspiracy (31.6%) (Figure 1).

Multivariate logistic regression was performed to identify the influencing factors of vaccination rejection, which were grouped under various constructs of SCT. Significant predictors in the construct of reciprocal determinism included nationality (aOR 0.03, 95% CI 0.00–0.60), history of COVID-19 (aOR 3.29, 95% CI: 1.08–9.98), an income of more than SAR 20,000 per month (aOR 10.13, 95% CI: 15–89.17), and television as the main source of information about COVID-19 (OR 3.40, 95% CI: 1.29–8.99). When considering behavior, being unsure about vaccine safety was associated with higher odds of vaccine rejection (aOR 4.92, 95% CI: 1.85–13.11). None of the variables included in the expectancy construct were found to be significantly associated with vaccine rejection. In the self-efficacy category, those who did not register for vaccination were more likely to reject the vaccine. Observational learning was a significant predictor of vaccination, as those who did not follow friends and other family members were more than four times more likely to reject the vaccine (aOR 4.12, 95% CI: 1.53–11.06). In the construct of reinforcement, there was no significant association of government enforcement with the decision of rejecting vaccines (Table 3).

## 4. Discussion

In this study we estimated the COVID-19 vaccine rejection rate and barriers to vaccination and attempted to explain these by using social cognitive theory. We found that nearly one-third of respondents rejected the vaccines. The belief that vaccine may not be effective was the most common reason for rejection. Nationality, history of getting COVID-19, if friends or family received the vaccine, and reinforcement were significant predictors for the vaccine rejection.

We found that the rejection rate was 30.5%, which is approximately consistent with the findings of another study done in Qassim. They collected data from March to May 2021 and reported that 14.7% refused vaccination and 22.7% were still undecided [25]. Rejection was higher in a study done by Magadmi et al. in Saudi Arabia, which reported a 55.3% rejection rate [6]. Additionally, two studies were done in Saudi Arabia that reported the difference between vaccine rejection before and after the vaccine was available. Before the roll out of the vaccines only 7% refused the vaccine and 28.2% were not sure [5]. After the vaccine was available 46.7% of the participants reported that they would take it only if it is mandatory [13]. On the other hand, varying rates of vaccine rejection have been reported in Middle Eastern countries. The rejection rate was reported to be 45.2% in Qatar [26], while another study, which included participants from all the Arab countries, showed a 38% rejection rate [9]. Earlier studies on COVID-19 vaccination related behaviors from different countries showed low rejection rates: China, 8.7% [18]; Israel, 20% [19]; and Hong Kong 17% [17]. Studies from low-income countries have generally reported higher rejection rates. For example, studies from Ghana and Ethiopia reported higher rejection rates of 46% and 37%, respectively [27,28]. The variation in the vaccine hesitancy across the studies could be due to the differences in time when the studies were conducted, the setting of the study, and the local burden of COVID-19 infections. Our finding of a 30% rejection rate should be considered as a high level of COVID-19 vaccine hesitancy, especially when it is freely available to all of the population and enforcement measures exist, such as making vaccination mandatory for entering holy places and shopping malls as well as for travelling outside the country.

It is important to explore barriers behind hesitancy and rejection of COVID-19 vaccines in order to overcome these issues and reach the goal of an immunized society. Our study showed that the most common barriers were believing the vaccines are not effective, not safe, and have serious side effects. Similarly, a study done in the Qassim region showed the reason was mainly that people did not believe in the vaccines [25]. Other studies done in Saudi Arabia showed consistent results with our study. One study showed that concerns about the side effects were the barrier for most refusers (80%), a lack of trust concerning the effectiveness of the vaccines was reported by 25%, and only 15% believed in a conspiracy theory about vaccines [6]. Another study, showed that around half of participants refused the vaccine because of concerns about the effectiveness and for 33.1% of participants the reason for rejection was information from social media sources [11]. Alobaidi S, in his study, reported that 48.1% of participants were worried about the safety of the vaccine, 45.2% expressed disbelief about the COVID-19 vaccine, and 42.6% were anxious about possible side effects [12]. The fast verification of the safety and effectiveness of the vaccine has been a major concern in many studies from Arab countries as well as in England [10,24]. This represents a big challenge for health authorities as the rapid evolution of the pandemic made the need for a vaccine inevitable. Effective mass education using contextual media preferences in specific countries are required to address the prevailing concerns among the population about the safety of the COVID-19 vaccines. Our study and other studies have reported on the influence of the media (of various types) on vaccine acceptance rates [26,29].

We found that non-Saudi people were less likely to reject the vaccine. This could be due to the fact that non-Saudis would need to be vaccinated to travel to their home countries. Having a previous COVID-19 infection was also associated with high vaccine rejection rates. We had excluded recently infected people because they were ineligible for COVID-19 vaccine in Saudi Arabia. This may indicate a perception among previously infected people of continuing immunity after infection. Educating people about disease and immunity patterns would help address this issue. We found a high income to be associated with higher odds of vaccine rejection. Other studies, however, have reported that lower income was associated with vaccine rejection. Another study from Saudi Arabia reported no association between income and vaccination acceptance [13]. In contrast to our findings, other studies have shown vaccine rejection to be associated with a low monthly income [24,30].

Observational learning is important in adopting something new. This is important in the case of the COVID-19 vaccination campaign as it was rolled out rapidly and people had concerns regarding its safety and efficacy. We found that observing others (friends and family members) was an influencing factor in vaccination decisions. This finding has practical implication as governments can use role models (religious personalities and celebrities) to influence health behaviors of the population for COVID-19 vaccination or for any other disease.

Surprisingly, we did not find government enforcement as significant predictor of vaccine hesitancy. There was a significant increase in the proportion of people accepting vaccination in our data, but when we adjusted for other variables, government mandates became a non-significant variable. This might indicate that some health behaviors, such as COVID-19 vaccination, may not be influenced by enforcement, but rather require an effective informational and educational communication strategy.

We successfully applied SCT to explain COVID-19 vaccine rejection, which is among the few attempts to look at this issue from a social perspective. There are certain limitations that need to considered while interpreting the results of this study. Firstly, the participants were recruited conveniently from PHC centers of one city; therefore, the results may not be generalizable to all of Qassim region or all of Saudi Arabia. Nonetheless, our sample has better population representation than other studies where an online approach was used, which may exclude elderly and illiterate people. Furthermore, our sample is closer to the general population in some of the characteristics such as educational status (up to secondary level 35% vs. 33%) [31], economic participation (46% vs. 49.5%) [32], monthly income (SAR ~12,500 vs. 14,832) [33], and influenza vaccine coverage (29% vs. 37%) [34]. Secondly, this was a cross-sectional study that measured the vaccination intention at a single point in time. Rapid evolution of the pandemic, population awareness, and other factors might affect this behavior. Thirdly, there could be a possibility of social desirability bias in reporting knowledge and preferences. However, this may have little effects on the validity of our results as we collected anonymized the data through a self-administered questionnaire. Lastly, our sample was powered for COVID-19 vaccine rejection only; therefore, it may not have enough power for all the associations we observed. This is also evident from the wide confidence intervals. Nevertheless, we have explored various individual, social, and environmental factors associated with vaccination behavior.

## 5. Conclusions

Nearly one-third of the participants showed vaccine hesitancy and the most common reason behind vaccine hesitancy was a concern about vaccine safety and effectiveness. This should steer policy makers to develop effective mass health education interventions in order to address disbeliefs and myths related to COVID-19 vaccines among the population by using common channels such as TV/radio and social media. Social and environmental factors also play an important role in modelling vaccination behavior, along with other individual factors. These factors need to be studied and addressed contextually by using health promotion theories such as social cognitive theory.

## Figures and Tables

**Figure 1 vaccines-09-01304-f001:**
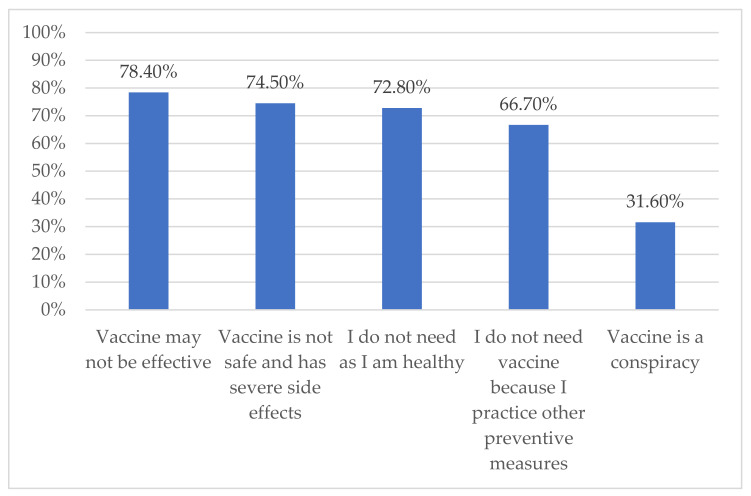
Reasons for rejecting COVID-19 vaccine.

**Table 1 vaccines-09-01304-t001:** Socio-demographic and health characteristics of the study population (*n* = 486).

Variable		% (*n*)
Age (*n* = 474)	Mean (SD)	32.80 (12.08)
Gender	Male	54 (262)
	Female	46 (223)
Marital status	Never married	46.1 (219)
	Ever Married	53.9 (256)
Nationality	Saudi	90.5 (428)
	Non-Saudi	9.5 (45)
Educational level	High school and lower	34.8 (165)
	Diploma and higher	65.2 (309)
Employment status	Unemployed	53.6 (254)
	Employee	46.4 (220)
Monthly Income	Less than 5000 SAR	46.6 (194)
	5001–10,000 SAR	26 (108)
	10,001–20,000 SAR	23.6 (98)
	More than 20,000 SAR	3.8 (16)
Any chronic disease	No	84.2 (405)
	Yes	15.8 (76)
Did you get influenza vaccine in last two years?	No	71.3 (338)
	Yes	28.7 (136)
Did you suffer from COVID-19?	No	84.1 (397)
	Yes	15.9 (75)
Did any of your family members suffer from COVID-19?	No	62.7 (296)
	Yes	37.3 (176)
In your opinion your risk of getting COVID-19 infection is	Less than 10%	33.3 (148)
	10–40%	39.8 (177)
	>40%	27(120)

**Table 2 vaccines-09-01304-t002:** Awareness and behaviors towards COVID-19 and its vaccines.

Variable	% (*n*)
COVID-19 disease could lead to death	
Yes	68.2 (324)
No	11.4 (54)
Not sure	20.4 (97)
COVID-19 may lead to hospitalization	
Yes	87.8 (417)
No	4.6 (22)
Not sure	7.6 (36)
COVID-19 spreads by close contact to infected people	
Yes	91.9 (434)
No	4 (19)
Not sure	4 (19)
COVID-19 can be prevented by precautionary measures	
Yes	84.3 (397)
No	6.4 (30)
Not sure	9.3 (44)
I have heard about COVID-19 vaccine	
Yes	97.7 (460)
No	1.7 (8)
Not sure	0.6 (3)
COVID-19 vaccine is available in the Kingdom	
Yes	93.8 (442)
No	1.9 (9)
Not sure	4.2 (20)
COVID-19 vaccine is effective in preventing the disease	
Yes	63.2 (299)
No	6.3 (30)
Not sure	30.4 (144)
COVID-19 vaccine is safe	
Yes	61 (288)
No	6.6 (31)
Not sure	32.4 (153)
If available, I will get COVID-19 vaccine for my family member	
Yes	76.2 (342)
No	23.8 (107)
I already registered for get COVID-19 vaccine	
Yes	50.9 (223)
No	49.1 (215)
I will get my vaccine only if required by my employer	
Yes	40.7 (174)
No	59.3 (253)
I will get COVID-19 vaccine only if my friends or other family members get it first	
Yes	28 (120)
No	72 (308)
I wash my hands frequently	
Yes	86.2 (405)
No	13.8 (65)
I use sanitizer frequently	
Yes	74.3 (349)
No	25.7 (121)
I always wear mask in public places	
Yes	90.9 (430)
No	9.1 (43)

**Table 3 vaccines-09-01304-t003:** Association of vaccine rejection with various constructs of social cognitive theory in Qassim, Saudi Arabia.

Variables	Univariable		Multivariable	
	OR (95% CI)	*p*-Value	OR (95% CI)	*p*-Value
Reciprocal determinism				
Gender Male Female	1 1.30 (0.88–1.92)	0.188	1 1.72 (0.75–3.93)	0.199
Marital status Never married Ever Married	1 1.13 (0.76–1.68)	0.550		
Nationality Saudi Non-Saudi	1 0.27 (0.10–0.69)	0.007	1 0.03 (0.00–0.60)	0.022
Educational level High school and lower Diploma and higher	1 1.12 (0.74–1.70)	0.599		
Employment status Unemployed Employee	1 0.95 (0.64–1.41)	0.799		
Monthly Income Less than SAR 5000 SAR 5001-10000 SAR 10001–20000 SAR More than 20000	1 1.10 (0.66–1.83) 0.80 (0.46–1.39) 1.02 (0.34–3.06)	0.706 0.424 0.976	1 1.26 (0.46–3.49) 2.24 (0.82–6.13) 10.13 (1.15–89.17)	0.653 0.117 0.037
Any chronic disease? No Yes	1 0.80 (0.45–1.40)	0.427		
Did you get influenza vaccine in last two years? No Yes	1 0.31 (0.19–0.53)	<0.001		
Did you suffer from COVID-19? No Yes	1 1.71 (1.03–2.86)	0.040	1 3.29 (1.08–9.98)	0.036
Did any of your family members suffer from COVID-19? No Yes	1 1.28 (0.86–1.92)	0.226	1 0.51 (0.19–1.33)	0.168
Television/radio No Yes	1 1.87 (1.24–2.81)	0.003	1 3.40 (1.29–8.99)	0.013
News papers No Yes	1 2.20 (1.35–3.59)	0.002	1 1.22 (0.39–3.81)	0.739
Social Media (e.g., Facebook, Twitter, Instagram, WhatsApp) No Yes	1 1.47 (0.98–2.23)	0.065	1 1.52 (0.42–5.47)	0.525
Friends and Family Members No Yes	1 1.35 (0.91–2.00)	0.134	1 1.33 (0.40–4.35)	0.641
Ministry of Health’s COVID-19 related information Yes No	1 2.01(1.35–3.01)	0.001	1 0.57 (0.16–2.00)	0.377
Health Care providers Yes No	1 1.92 (1.29–2.85)	0.001	1 0.54 (0.16–1.85)	0.328
Behavioral Capability				
COVID-19 spreads by close contact to infected people Yes No Not sure	1 2.25 (0.85–5.96) 4.34 (1.67–11.27)	0.104 0.003	1 4.54 (0.90–22.86) 2.36 (0.28–20.15)	0.066 0.432
COVID-19 can be prevented by precautionary measures Yes No Not sure	1 1.59 (0.73–3.48) 2.85 (1.52–5.36)	0.245 0.001	1 0.36 (0.06–2.02) 2.98 (0.91–9.76)	0.246 0.071
I have heard about COVID-19 vaccine Yes No Not sure	1 14.09 (1.68–118.15) 1.17 (0.11 -13.06)	0.015 0.896		
COVID-19 vaccine is available in the Kingdom Yes No Not sure	1 7.44 (1.48–37.36) 3.41 (1.34–8.68)	0.015 0.010		
COVID-19 vaccine is effective in preventing the disease Yes No Not sure	1 21.30 (8.21–55.22) 6.23 (3.94–9.86)	<0.001 <0.001	1 4.44 (0.80–24.62) 1.42 (0.50–4.04)	0.088 0.507
COVID-19 vaccine is safe Yes No Not sure	1 10.49 (4.65–23.69) 7.53 (4.73–11.97)	<0.001 <0.001	1 4.83 (0.80–29.15) 4.924 (1.85–13.11)	0.086 0.001
I wash my hands frequently Yes No	1 4.34 (2.50–7.54)	<0.001	1 2.84 (0.89–9.10)	0.078
I use sanitizer frequently Yes No	1 2.65 (1.71– 4.11)	<0.001		
I always wear mask in public places Yes No	1 2.62 (1.37– 5.01)	0.004		
Expectations				
In your opinion your risk of getting COVID-19 infection is Less than 10% 10–40% >40%	1 0.92 (0.58–1.47) 0.54 (0.31–0.95)	0.723 0.031		
COVID-19 could lead to death Yes No Not sure	1 1.90(1.03–3.53) 2.20 (1.37–3.54)	0.041 0.001		
COVID-19 may lead to hospitalization Yes No Not sure	1 2.26 (0.90–5.71) 2.01 (1.01–4.01)	0.084 0.048		
Self-efficacy				
If available, I will get COVID-19 vaccine for my family member Yes No	1 137.20(61.96–303.79)	<0.001		
I already registered for get COVID-19 vaccine Yes No	1 15.86(8.52–29.50)	<0.001	1 20.80 (6.99–61.87)	<0.001
Observational learning				
I will get COVID-19 vaccine only if my friends or other family members get it first Yes No	1 1.02 (0.64–1.62)	0.947	1 4.12 (1.53–11.06)	0.005
Reinforcements				
I will get my vaccine only if required by my employer Yes No	1 0.37(0.24–0.56)	<0.001	1 0.20 (0.09–0.46)	<0.001
Period Pre enforcement Post enforcement	1 0.48 (0.33–0.72)	<0.001		

## Data Availability

Data used in this study can be obtained from corresponding author on request.

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
