# Peer review of "Explaining COVID-19 Vaccine Rejection Using Social Cognitive Theory in Qassim, Saudi Arabia"

_vaccines, 2021, doi:10.3390/vaccines9111304_

Round 1

Reviewer 1 Report

the article from Al Saeed et al, is an interesting study that evaluated the reasons of COVID-19 vaccine rejection in Saudi Arabia. The authors utilize social cognitive theory and conduct logistic regression analysis to identify the main reason for vaccine denial. Minor corrections are required.

  • page 3, line 96, "within each selected PHC, a total of 40-50 participants were selected consecutively by convenience sampling". please specify
  • please avoid the word " About" in the results session
  • The article was many figures and tables and some info can be mentioned in text onlt. The authors may consider remove Figure 1 and 2 and descibe results in the text n the results section.

Author Response

We are thankful to the reviewers for their time efforts to identify the areas of improvement in the manuscript. We have thoroughly gone through the comments and revised the manuscript as per comments. Following is the point-by-point response to each of the comments from the reviewers. All the changes in the manuscript have been highlighted yellow.

The article from Al Saeed et al, is an interesting study that evaluated the reasons of COVID-19 vaccine rejection in Saudi Arabia. The authors utilize social cognitive theory and conduct logistic regression analysis to identify the main reason for vaccine denial. Minor corrections are required.

page 3, line 96, "within each selected PHC, a total of 40-50 participants were selected consecutively by convenience sampling". please specify 

Response: We assume that giving specific number of participants from each of the PHC would not add much to the understanding of methods of this study and its validity as there was not much variations in the numbers except one PHC where we could get only 40 participants. Nevertheless, following is the detail of numbers from each of the PHC (50, 50, 50, 50, 50, 50, 49, 49, 48, 40).

please avoid the word " About" in the results session

Response: We have revised and edited as per suggestion.

The article was many figures and tables and some info can be mentioned in text onlt. The authors may consider remove Figure 1 and 2 and descibe results in the text n the results section.

Response: As suggested, we have removed figures 1 and 2, and described in the results’ description.

Reviewer 2 Report

The research work carried out by the authors is very interesting and innovative. The research is well developed. The conclusions are rather flat. The authors should make a large change in the conclusions to get the manuscript fit for publication. There are grammatical errors throughout the manuscript.

Author Response

We are thankful to the reviewers for their time efforts to identify the areas of improvement in the manuscript. We have thoroughly gone through the comments and revised the manuscript as per comments. Following is the point-by-point response to each of the comments from the reviewers. All the changes in the manuscript have been highlighted yellow.

The research work carried out by the authors is very interesting and innovative. The research is well developed. The conclusions are rather flat. The authors should make a large change in the conclusions to get the manuscript fit for publication. There are grammatical errors throughout the manuscript.

Response: Thank you so much for your valuable comments. We have revised the conclusion and revised the draft for grammatical errors.

Reviewer 3 Report

In this article, the authors elaborate on Vaccine hesitancy and rejection towards the SARS-CoV-2  Vaccination on the basis of a survey administered in Qassim (Saudi Arabia) and comment it using Social Cognitive Theory.

The title of the article is straightforward and apposit.
The abstract is and appropriate in lengh and content, but the last sentences should be put near the beginning of the abstract (because they illustrate the reasons behinf the study), leaving at the end of the abstract the conclusions.. 
The keywords may be improved (I suggest to change "Acceptance" in "Vaccine acceptance", add "Vaccine hesitancy" and remove (or improve) "Barriers" (that as a single word is not so useful as Keyword, and maybe even a bit misleading)
The introdution is more than sufficient to provide the necessary background.
The references are relevant and recent (please fill the "accessed on" blank spaces)

The survey and data collection procedure description is clear and complete enough.
The Socio-demographic and health characteristics of the study population is clearly described, but it may be of interest to quickly compare it with the average population of the country, to show how well it reflects it (or how much it has to be normalized to reflect the country's population).
Tables readibility may be improved separating the different questions with horizontal lines. In Table 1 I also suggest to put the characteristic in a first column, the options in a second column and the % in the third one; also pleas clarify the Age section. Figures are clearly readable and well labeled.

The discussion of data il clear and pertinent and the conclusions seem consistent with the experimental data. Maybe some comment coud be devoted to the known limitations of the study.
There are some English imperfections, in particular in the use of punctuation.
The scientific soundness and significance of content are average, the originality is not so high, but anyway this study contribute to enlighten such an important social and health issue, helping to individuate the possible targets to an informative and communicative campaign. Also for this reason, it may be of potential interest for the readers of the journal, and so in my opinion it is, all considered, suited for publication (after suggested editings).

Author Response

We are thankful to the reviewers for their time efforts to identify the areas of improvement in the manuscript. We have thoroughly gone through the comments and revised the manuscript as per comments. Following is the point-by-point response to each of the comments from the reviewers. All the changes in the manuscript have been highlighted yellow.

In this article, the authors elaborate on Vaccine hesitancy and rejection towards the SARS-CoV-2  Vaccination on the basis of a survey administered in Qassim (Saudi Arabia) and comment it using Social Cognitive Theory.

The title of the article is straightforward and apposit.

Response: Thanks for the compliment.
The abstract is and appropriate in lengh and content, but the last sentences should be put near the beginning of the abstract (because they illustrate the reasons behinf the study), leaving at the end of the abstract the conclusions.. 

Response: Thanks for suggestion. However, the last sentence in the abstract is one of our conclusions that social cognitive theory can be used to understand the vaccination behavior.

The keywords may be improved (I suggest to change "Acceptance" in "Vaccine acceptance", add "Vaccine hesitancy" and remove (or improve) "Barriers" (that as a single word is not so useful as Keyword, and maybe even a bit misleading)

Response: Thanks for the suggestion. We have revised keywords as suggested.

The introdution is more than sufficient to provide the necessary background.
The references are relevant and recent (please fill the "accessed on" blank spaces)

Response: References have been corrected.

The survey and data collection procedure description is clear and complete enough.
The Socio-demographic and health characteristics of the study population is clearly described, but it may be of interest to quickly compare it with the average population of the country, to show how well it reflects it (or how much it has to be normalized to reflect the country's population).

Response: Thanks for the suggestion. We have already commented on representativeness of our sample as compared to other studies used online approach. Nevertheless, we have further added the comparison of our sample with general population. Page 11; lines: 264-272.

Tables readibility may be improved separating the different questions with horizontal lines. In Table 1 I also suggest to put the characteristic in a first column, the options in a second column and the % in the third one; also pleas clarify the Age section. Figures are clearly readable and well labeled.

Response: Thanks for the suggestion. We have added horizontal lines to separate the variables within tables. Table 1 formatted as suggested.  

The discussion of data il clear and pertinent and the conclusions seem consistent with the experimental data. Maybe some comment coud be devoted to the known limitations of the study.

Response:  Limitations sections has been elaborated further.

There are some English imperfections, in particular in the use of punctuation.

Response: We have revised the whole draft for such errors and corrected accordingly.

The scientific soundness and significance of content are average, the originality is not so high, but anyway this study contribute to enlighten such an important social and health issue, helping to individuate the possible targets to an informative and communicative campaign. Also for this reason, it may be of potential interest for the readers of the journal, and so in my opinion it is, all considered, suited for publication (after suggested editings).